# Deep Multi-Branch CNN Architecture for Early Alzheimer’s Detection from Brain MRIs

**DOI:** 10.3390/s23198192

**Published:** 2023-09-30

**Authors:** Paul K. Mandal, Rakeshkumar V. Mahto

**Affiliations:** 1Department of Computer Science, University of Texas, Austin, TX 78712, USA; 2Department of Electrical and Computer Engineering, California State University, Fullerton, CA 92831, USA; ramahto@fullerton.edu

**Keywords:** Alzheimer’s, brain imaging, CNN, convolution, convolutional neural network, deep learning, disease detection, neural network, machine learning, medical diagnosis

## Abstract

Alzheimer’s disease (AD) is a neurodegenerative disease that can cause dementia and result in a severe reduction in brain function, inhibiting simple tasks, especially if no preventative care is taken. Over 1 in 9 Americans suffer from AD-induced dementia, and unpaid care for people with AD-related dementia is valued at USD 271.6 billion. Hence, various approaches have been developed for early AD diagnosis to prevent its further progression. In this paper, we first review other approaches that could be used for the early detection of AD. We then give an overview of our dataset and propose a deep convolutional neural network (CNN) architecture consisting of 7,866,819 parameters. This model comprises three different convolutional branches, each having a different length. Each branch is comprised of different kernel sizes. This model can predict whether a patient is non-demented, mild-demented, or moderately demented with a 99.05% three-class accuracy. In summary, the deep CNN model demonstrated exceptional accuracy in the early diagnosis of AD, offering a significant advancement in the field and the potential to improve patient care.

## 1. Introduction

Alzheimer’s disease (AD) is a common disease that affects 1 in 9 (10.7%) Americans over 65. Six and a half million Americans aged 65 or over have been diagnosed with AD dementia. An estimated 16 billion unpaid hours of care were given to people with dementia from AD in 2021, representing an estimated value of USD 271.6 billion [1]. Given the substantial impact of AD on individuals and society, it is essential to understand its progressive nature, which can be systematically classified into distinct stages [2].

These stages encompass the early or preclinical phase, characterized by subtle or absent symptoms, followed by mild cognitive impairment (MCI), which is marked by noticeable memory problems while maintaining independence. Subsequently, individuals may transition into the mild dementia stage, characterized by difficulties with memory, daily tasks, communication, and mood fluctuations. Further progression leads to moderate dementia, where daily assistance becomes necessary, and the recognition of loved ones becomes increasingly challenging. In the advanced stages, known as severe dementia, individuals rely entirely on caregivers for support and often experience substantial difficulties in communication and recognition. Approximately 12% to 18% of people over 60 are living with MCI [3]. MCI causes subtle changes in memory and thinking. Although often associated with the normal aging process, MCI is not a part of typical aging. Moreover, 10–15% of individuals with MCI develop full dementia each year [4]. Therefore, AD must be diagnosed at an early stage to prevent it from progressing further.

For this purpose, machine learning (ML) and deep learning (DL) can play an invaluable role, since they have been extensively used in various other medical applications for diagnosing and detecting various abnormalities and diseases [5,6,7]. A diverse range of approaches have been employed in the field of early AD detection, encompassing the analysis of speech patterns and inflections, neuropsychometric tests, olfactory tests, eye testing, gait testing, and the utilization of neural networks for various diagnostic modalities such as MRIs, electroencephalograms (EEGs), and magnetoencephalographs (MEGs) [8]. Recently, there has been a surge in the popularity of ML and DL techniques for early AD diagnosis, with a predominant focus on applying these methods to MRI images, MEGs, EEGs, and other relevant physiological parameters [9]. However, it is worth noting that the accuracy achieved in each of these techniques falls short of 99%. Hence, there is a pressing need for the development of new architectural approaches to further enhance AD diagnosis through the utilization of ML and artificial intelligence (AI).

In this research endeavor, we embarked on an in-depth exploration of recent advancements in the application of deep learning for the detection of AD. Our study not only involved a meticulous examination of the current state of the field but also sought to address crucial limitations that exist within this domain. Our investigation revolved around a novel deep convolutional neural network (CNN) architecture, notable for its substantial parameter count, totaling 7,866,819. This architectural innovation introduces three distinct convolutional branches, each varying in length and incorporating different kernel sizes. This technique allows for better and more nuanced feature extraction. The primary novelty we present in this paper centers on the conception and rigorous evaluation of this intricate CNN model, with the ultimate objective of advancing the accuracy and robustness of AD detection.

The paper is organized as follows: Section 2 details prior research into AI/ML methods for AD detection, with a specific focus on their utilization for the analysis of MRI images, MEGs, EEG data, and other pertinent physiological parameters. Section 3 provides an in-depth description of the dataset employed in this study, offering insights into its composition and relevance. The methodology employed for training the CNN model is detailed in Section 4. Section 5 offers a comprehensive presentation of the experimental results and an in-depth discussion of their implications. Finally, the paper concludes in the last section, summarizing the key findings and their significance in the context of AD diagnosis while also pointing toward future research directions.

## 2. Related Work

In AD diagnosis, ML and AI methodologies have played pivotal roles. Notably, prior research in this domain has underscored the application of these techniques for analyzing diverse data modalities, including MRI images, MEG recordings, EEG data, and various pertinent physiological parameters.

### 2.1. Neural Networks for MRIs

Various neural network techniques have been used to predict AD. In [10], a convolution neural network (CNN) was applied to accurately predict mild cognitive impairment in AD. The overall accuracy reported in [10] was around 86.1%. A similar CNN technique was applied to a dataset consisting of MRI images of 156 AD and 156 normal patients [11]. The dataset in this study consisted of AD patients and age/gender-matched normal individuals [11]. The technique proposed in [11] achieved an accuracy of 94%. A mix of a 2D CNN and recurrent neural network (RNN) for MRI images was reported to achieve an accuracy of 96.88% [12]. The proposed technique applied an RNN after applying a 2D CNN to recognize the connection between 2D image slices [12]. The study also presented a technique for transferring learning from 2D images to 3D CNNs.

Over the last 12 years, there has been a comprehensive exploration of diverse techniques employed in diagnosing AD through the application of AI methodologies to MRI images. A thorough analysis and assessment of these techniques was presented in the work by Frizzell et al. [13], shedding light on the advancements and innovations that have taken place in this dynamic field.

### 2.2. Neural Networks for Magnetoencephalographs (MEGs)

In contrast to MRI images, a non-invasive diagnostic technique called magnetoencephalography (MEG) is utilized for measuring brain activity. Based on brain activity, the proposed method estimates the magnetic field generated by the slow ionic current flow through cells. Research has shown that MEG activity can provide excellent sensitivity for the early diagnosis of DP [14]. A combination of MEG recordings and MRI scans were utilized in [15], which resulted in an accuracy of 89%. A similar technique for diagnosing AD was presented in [16]. However, the accuracy of the classification technique was 77%. Various other ML-driven techniques for diagnosing AD using MEGs were summarized in [17]. However, none of the techniques were able to achieve an accuracy greater than 90%.

### 2.3. Neural Networks for Electroencephalograms (EEGs)

Another more promising study used electroencephalograms to detect AD. Electrophysiological imaging techniques such as EEGs are widely accepted as reliable indicators for the diagnosis of AD. With the aid of neural networks, it has become possible to use EEG data to accurately determine whether a patient has AD. A novel neural network, I-Fast, was able to predict whether subjects had AD with 92% accuracy [18]. The dataset used in this study consisted of 115 mild cognitive impairment and 180 AD patients. This is significant as it implies that EEGs can be used as a viable alternative for the diagnosis of AD, given the cost-effectiveness of the technique. Similarly, a novel technique was presented in [19] that used a finite response filter (FIR) in a double time domain to extract features from an EEG recording dataset consisting of MCI, AD, and healthy control (HC) subjects. Later, binary classification (BC) achieved an accuracy of 97%, 95%, and 83% for HC vs. AD, HC vs. MCI, and MCI vs. AD, respectively.

### 2.4. Blood Plasma

The development of a molecular diagnostic test for AD holds considerable potential for advancing therapeutic interventions. In the scope of the investigation presented in [20], the authors successfully identified a panel of 18 signaling proteins within blood plasma, demonstrating the ability to discriminate samples from individuals with AD and those from control subjects with a remarkable accuracy of nearly 90%. Furthermore, these proteins exhibited promising utility in the early identification of individuals with mild cognitive impairment who subsequently progressed to AD within a 2–6 year timeframe [20]. The comprehensive biological analysis of these identified proteins revealed a systemic dysregulation encompassing processes such as hematopoiesis, immune responses, apoptosis, and neuronal support during the presymptomatic stages of AD [20]. Leveraging a dataset comprising 259 archived plasma samples collected across a spectrum from presymptomatic to late-stage AD, alongside control subjects, the findings in [20] offered significant potential for enhancing the comprehension and diagnostic capabilities related to AD. Additionally, this is probably the most promising of the methods described above, since it is much easier and less costly to run blood tests.

In addition to blood-based approaches, in a separate study, the levels of AD-related proteins in plasma neuronal-derived exosomes (NDEs) were quantified to identify biomarkers for the prediction and staging of MCI and AD [21]. This research revealed that abnormal levels of specific proteins within plasma NDEs, such as P-tau, Aβ1-42, NRGN, and REST, accurately predicted the conversion of MCI to AD dementia and induced AD-like neuropathology when injected into the central nervous system of normal mice, highlighting their potential as crucial diagnostic indicators [21]. At the prodromal stage and with even stronger performance in the advanced stages of the disease, disease detection models utilizing the identified panels demonstrated a sensitivity (SN) exceeding 80%, specificity (SP) surpassing 70%, and area under the receiver operating curve (AUC) of at least 0.80 [21]. These techniques offer simpler methods for early AD diagnosis, though their accuracy pales in comparison to applying ML/AI to MRI images.

### 2.5. Other Techniques

One of the best-performing models that did not rely on MRIs was a neural network trained to analyze speech patterns. One such model was reported to have a 97.18% accuracy [22]. However, there were two main issues with this approach. First and foremost, it is clear from looking at the audio waves that the subjects who had AD were well past the MCI/mild demented stage, making it non-viable for an early detection stage. The second was that the study only included 50 non-demented subjects and 20 demented subjects. Each non-demented subject provided 12 h of audio, and each demented subject provided 8 h of audio. These clips were divided into 600 different clips comprising 60 s of audio. However, the authors did not state whether they divided the training and validation sets by patient. If this were the case, there would be a possibility that the neural network learned how to classify whether the subject had AD based on the patient’s voice rather than extracting useful information.

Similarly, in a recent study, researchers harnessed the potential of non-invasive biomarker discovery in AD using saliva as a biofluid [23]. This study employed a comprehensive metabolomics workflow, differentiating CN, MCI, and other AD groups. This groundbreaking approach, which included the identification of distinctive biomarkers and diagnostic panels, showcased impressive diagnostic accuracy in the discrimination of AD from CN and MCI [23]. This suggests that saliva holds significant promise as a biofluid for advancing global AD biomarker research. Similarly, another study utilizing machine learning, including neural networks and information theoretic feature selection, achieved a diagnostic accuracy exceeding 83% in classifying MCI subtypes and AD based on digital clock drawing test (dCDT) data [24]. This demonstrates the potential of AI/ML to detect neurodegenerative diseases without using MRIs [24]. It is noteworthy, however, that despite the advancements in utilizing ML and AD for Alzheimer’s diagnosis, the accuracy achieved with these methods still lags behind that attainable when applied to MRI images.

## 3. Description of Alzheimer’s MRI Datasets

In this study, we leveraged the rigorously curated Alzheimer’s dataset, cited as [25], which consists of magnetic resonance imaging (MRI) scans. These scans are systematically classified into four diagnostic categories: mild demented, moderate demented, non-demented, and very mild demented. The dataset is a robust platform for training and evaluating deep learning algorithms targeted at the accurate staging of AD. Accessible on Kaggle, this dataset not only supports the research community by facilitating algorithmic advancements in the diagnosis and treatment of AD but also contributes to alleviating the growing global burden of the disease.

We selected this dataset for its multiple advantages: it is freely accessible, offers a variety of diagnostic classifications, and requires minimal storage capacity, distinguishing it from other widely used datasets in the field. The dataset is an aggregation of 6338 MRI scans. These scans underwent preprocessing and curation before being made publicly available on Kaggle [26]. For standardization, the MRI scans were resized to a uniform resolution of 100 × 100 pixels. The dataset is partitioned into 3202 non-demented, 2242 very mild demented, and 892 mild demented scans. We intentionally excluded the moderate demented category (*n* = 64) from our analyses to mitigate the influence of sample size bias. Figure 1 presents representative scans from each category within the dataset, and Figure 2 delineates the dataset’s categorical distribution.

We divided our training and test sets into 5701 training images (2881 non-demented, 2017 very mild demented, and 802 mild demented) and 637 test images (321 non-demented, 225 very mild demented, and 90 mild demented), as shown in Figure 3. We used stratified random sub-sampling to ensure that the training and test sets had the same ratio of non-demented, very mild demented, and mild demented images. As opposed to our test set, which was set aside before we started any training or hyperparameter tuning, we generated our validation dataset using stratified random sub-sampling in order to select 637 images from the entire training set.

## 4. Methods

The primary objective of this study was to construct an effective and robust neural network model for the early diagnosis of AD. To achieve this, we implemented a multi-branch CNN using Keras [27], which is well-integrated with TensorFlow [28]. The decision to employ a multi-branch architecture was informed by its superior ability to capture hierarchical features in complex data, as has been demonstrated in the existing literature [29].

### 4.1. Computational Environment

The model was trained and evaluated on a computational system equipped with an NVIDIA GeForce RTX 2080S graphics card, NVIDIA Corporate, Santa Clara, CA, USA, which had 8GB of VRAM. The system was further augmented with an Intel Xeon W-10855M processor featuring 24 logical cores and 12 physical cores, complemented by 64 GB of main memory. This hardware configuration was selected to ensure efficient memory management and computational power, crucial for neural network training and evaluation.

### 4.2. Training Protocol

For training, the neural network was subjected to approximately 100 epochs. Each epoch took an average of 2 min, resulting in a total training time of around 3–4 h. Optimization was performed using the Adam optimizer, with the learning rate set to 0.001. This learning rate was selected based on preliminary experiments showing its effectiveness in achieving fast yet stable convergence.

### 4.3. Layers and Hyperparameters

#### 4.3.1. Input Layer

The input layer accepted a normalized 100 × 100 × 3 tensor. Raw image dimensions of 100 × 100 pixels and a color depth of 3 channels (RGB) were utilized. Pixel intensities, originally in the range of 0–255, were normalized to a [0, 1] range to ensure data consistency and faster convergence. A detailed description of the layers comprising the neural network architecture is offered, accompanied by an enumeration of the associated parameters. The architecture under consideration is graphically represented in Figure 4.

#### 4.3.2. Convolutional Layer

We defined the convolution operation, ∗, between two discrete functions f[m,n] and g[m,n] as
(1)$$\left(f*g\right)\left[m,n\right]=\sum_{i=-\infty }^{\infty }\sum_{j=-\infty }^{\infty }f\left[m,n\right]g\left[m-i,n-j\right]$$

Convolution is a useful operation in signal processing. Convolution can apply filters to images that can result in different effects, such as sharpening or smoothing [30]. In a neural network, the convolutional layers have several parameters: the number of filters; the kernel size, which determines the size of each filter; and the stride size, which determines how many pixels to “skip” before applying the filter to the next block of pixels.

Assuming a stride size of 1, for a given N×M image, we applied an i×j filter. We divided our image into (N−i+1)×(M−j+1) sub-images and performed an inner product between the filter and the sub-image. The result of the convolution was a filtered image with (N−i)×(M−j) pixels.

A convolutional network learns optimal filters via training filter parameters by minimizing the loss [31]. These trained filters can learn useful filters that can help subsequent layers classify data. This is applicable to many sorts of problems, such as facial recognition [32].

#### 4.3.3. Pooling

Pooling is a method of reducing the dimensions for the outputs of convolutional layers. This has two main advantages. First, it helps prevent convolutional networks from overfitting. Secondly, if applied properly, it can decrease the computational expense of a CNN. For this reason, pooling is often used in neural networks for mobile systems [33].

There are two main kinds of pooling, max pooling and average pooling. Max pooling takes the largest value in a kernel [34]. Average pooling instead takes the average of all of the outputs within a kernel [35]. Using average pooling results in a smoother output compared to max pooling.

#### 4.3.4. Convolutional Block

The architecture featured a specialized convolutional block comprising three distinct layers: a 2D convolutional layer, an average pooling layer, and a dropout layer for regularization. The convolutional layer used varied numbers of filters and kernel sizes depending on its location within the multi-branch structure. The average pooling layer was used in lieu of max pooling to retain more feature information, based on findings by [36]. The dropout layer was set to randomly deactivate a specific percentage of neurons during each training epoch to mitigate overfitting, a design choice supported by [37].

#### 4.3.5. Dense Layer

The dense (fully connected) layer served as a pivotal component in our neural network. It consisted of neurons that were fully interconnected with all neurons from the preceding layer, facilitating efficient information integration and enabling the model to recognize more complex data patterns.

### 4.4. Model Architecture

The proposed multi-branch architecture was composed of three separate branches, each designed to capture different levels of abstractions, as shown in Figure 4. The first branch incorporated convolutional layers with a 3 × 3 kernel size and an average pooling size of 2 × 2. Five convolutional blocks were instantiated, generating 32, 64, 128, 256, and 512 filters, respectively. Post-convolution, the data were flattened and subjected to a 50% dropout.

The second branch employed convolutional layers with a 5 × 5 kernel size and an average pooling size of 3 × 3. This branch contained three convolutional blocks generating 128, 256, and 512 filters. After convolution, the data underwent flattening, and a 50% dropout rate was applied.

The third branch featured convolutional layers with a 7 × 7 kernel size and an average pooling size of 5 × 5. Two convolutional blocks were included, generating 128 and 256 filters, respectively. Following the convolution process, the data were flattened, and a dropout rate of 50% was applied.

The outputs from these branches were concatenated into a single layer, which was then fed into a feed-forward neural network with two dense layers of 256 and 128 outputs, respectively, each followed by a 50% dropout. The network concluded with a softmax activation layer consisting of three neurons, each representing a class label: non-demented, very mild demented, and mild demented.

## 5. Experimental Results and Discussion

Figure 5 shows the performance of our model against the validation set. The metrics plotted include accuracy, precision, recall, and area under the curve (AUC), each providing a comprehensive view of the model’s diagnostic prowess. These metrics are widely used in ML and classification tasks to assess the performance of models. Each metric serves a specific purpose and provides valuable insights into different aspects of a model’s predictive capabilities. The *x*-axis represents the number of training epochs, capturing the iterative optimization process. As a model iteratively learns from the data, its diagnostic capabilities evolve, and the corresponding performance metrics showcase this progression. The *y*-axis represents the percentage value of each metric, reflecting the proportion of correct predictions and successful classification. The presented metrics offer distinct perspectives on the model’s performance:

### 5.1. Accuracy

Accuracy measures the ratio of correctly predicted instances to the total number of instances in the dataset. It provides an overall assessment of the model’s performance.
(2)$$\mathrm{Accuracy}=\frac{TP+TN}{TP+TN+FP+FN}$$

The term true positive (*TP*) refers to instances where the model accurately identified individuals who truly had AD, marking a correct positive prediction. True negative (*TN*) indicates situations where the model correctly identified individuals without AD as not having the condition. On the other hand, a false positive (*FP*) arose when the model incorrectly identified an individual as having AD when they did not, thus making an inaccurate positive prediction. Similarly, a false negative (*FN*) occurred when the model incorrectly predicted an individual as not having AD when they actually did.

### 5.2. Precision

Precision represents the proportion of true-positive predictions among all instances predicted as positive. It quantifies how well the model predicted positive cases when it made a positive prediction.
(3)$$\mathrm{Precision}=\frac{TP}{TP+FP}$$

### 5.3. Recall (Sensitivity)

Recall calculates the proportion of true-positive predictions among all actual positive instances. It assesses the model’s ability to identify all positive cases.
(4)$$\mathrm{Recall}=\frac{TP}{TP+FN}$$

### 5.4. AUC (Area under the Curve)

The AUC is used to evaluate the performance of a classification model’s ability to distinguish between positive and negative classes across various threshold settings. It represents the area under the receiver operating characteristic (ROC) curve. The ROC curve is a plot of the true-positive rate (recall) against the false-positive rate (1—specificity) at different threshold values. The AUC quantifies the overall discriminative power of the model.

### 5.5. Validation Loss

Validation loss is another crucial metric in ML and DL, serving as a fundamental indicator of a model’s performance during training and validation. It quantifies the disparity between a model’s predictions and the true target values on a validation dataset. Mathematically, validation loss can be defined as follows:(5)$$\mathrm{Validation}\;\mathrm{Loss}=\frac{1}{N}\sum_{i=1}^{N}{\left(y_{\mathrm{true}}^{\left(i\right)}-y_{\mathrm{pred}}^{\left(i\right)}\right)}^{2}$$
here:Validation Loss is the computed loss in the validation dataset.*N* is the number of samples in the validation dataset.$y_{\mathrm{true}}^{\left(i\right)}$ represents the true target value of the *i*-th sample.$y_{\mathrm{pred}}^{\left(i\right)}$ represents the model’s predicted value for the i-th sample.

The goal during model training is to minimize the validation loss. Lower values of validation loss indicate that the model’s predictions align more closely with the actual target values, signifying improved model performance.

In Figure 6, we present the validation loss plotted against the number of training epochs. This visualization provides insights into the model’s learning progress over time, intending to achieve lower validation loss values as training proceeds. It is noteworthy that by 100 epochs, the validation loss became significantly smaller, indicating substantial improvement in the model’s performance, as shown in Figure 6.

Our neural network achieved an accuracy of 99.05% as depicted in the confusion matrix shown in Figure 7a,b. Of our 637 test images, only 6 were misclassified, with all of these images belonging to the non-demented and very mild demented classes, as shown in Figure 7a,b. We found that our neural network did not overfit and that the validation accuracy ceased to improve after about 100 epochs. However, increasing the size of our model did not yield any increase in accuracy.

To further contextualize the performance and robustness of our proposed deep CNN model, we compared it with other established techniques in the field. The analysis in Table 1 demonstrates the effectiveness of the proposed AI-powered technique for diagnosing AD through MRI images. It serves as a critical comparison with previously published models, highlighting the robustness and potential of this approach.

In the landscape of AI-driven AD diagnosis, the judicious selection and utilization of MRI images has proven to be instrumental in enhancing accuracy and reliability. The provided table ingeniously encapsulates the essence of these advancements by distilling them into crucial components. Notably, it meticulously juxtaposes the proposed technique against a series of other models, taking into account critical factors such as dataset size, the sophistication of the employed techniques, and the resulting performance metrics.

As the numbers within Table 1 reveal, the proposed technique surpassed its contemporaries in terms of accuracy, signaling a breakthrough that could revolutionize the landscape of AD diagnosis. It emerges as a beacon of hope for accurate and early diagnosis, potentially leading to improved patient outcomes and a deeper understanding of the disease’s progression. The comprehensive nature of the comparison underscores the significance of the advancement, affirming that the proposed technique is positioned as a frontrunner in the field of AI-assisted AD diagnosis using MRI images.

Despite its exceptionally high accuracy, there are many steps that we could take to make our model more applicable to a clinical setting. The first would be to create a new method for preprocessing the Alzheimer’s data directly from ADNI [25] rather than using the possibly outdated data uploaded to Kaggle. It would also be useful to augment the data from the Open Access Series of Imaging Studies (OASIS) [38]. Attempting to preprocess our data in a way that is as agnostic to MRI machines as possible could potentially increase the model’s utility in the medical community.

## 6. Conclusions

The early diagnosis of AD is a critical challenge facing society today. The disease not only results in a severe reduction in brain function but also puts a significant burden on caregivers and the healthcare system. The results of this study demonstrate the potential of using deep learning techniques, specifically CNNs, to improve the early diagnosis of AD. The proposed architecture achieved a high accuracy of 99.05% in predicting the progression of the disease and is a step forward in developing tools for the early detection and management of AD. The high accuracy rate suggests that the model could serve as a reliable diagnostic tool, thereby facilitating timely intervention and potentially mitigating the progression of the disease. This, in turn, could alleviate the substantial emotional and economic burdens placed on patients, caregivers, and healthcare systems.

However, the current approach has its limitations, and several research directions could be pursued to make it more practical and valuable in a clinical setting. For example, it would be beneficial to improve the preprocessing methods across a more general dataset of MRI images to ensure that the data more closely fit the patient population. Additionally, a further exploration of the classifier’s performance on different MRI layers could provide valuable insights into disease progression. One promising avenue for future work is to implement a neural expert system that inputs each layer of the MRI into a separate branch of a network and concatenates the outputs for further analysis. Another possibility is to use a fully convolutional network to highlight areas of the brain that are indicative of AD, which could provide important information to clinicians in making a diagnosis.

Additionally, more complex models could be created that combine our neural network architecture with the other methods enumerated in the Previous Research section in order to achieve classifiers with an even higher accuracy or, alternatively, factor in demographic data using a multi-modal approach. One field of inquiry that could be useful is the construction of a federated expert system including a heterogeneous federated ML model designed so that each “expert” trains on a different federate, but such a project is well outside the scope of this paper.

## Figures and Tables

**Figure 1 sensors-23-08192-f001:**
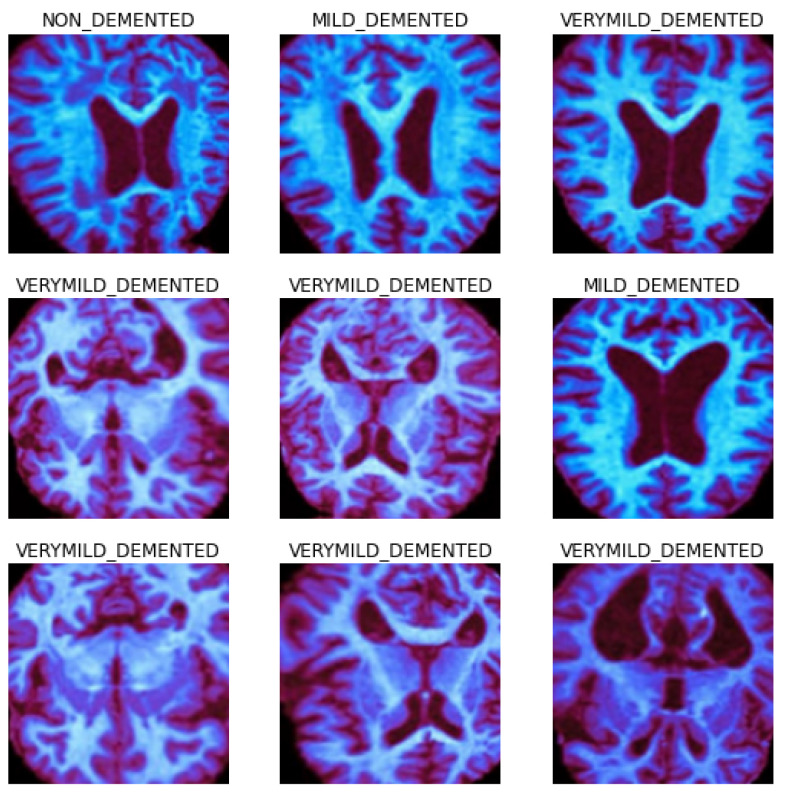
A sample of 9 preprocessed images from our dataset.

**Figure 2 sensors-23-08192-f002:**
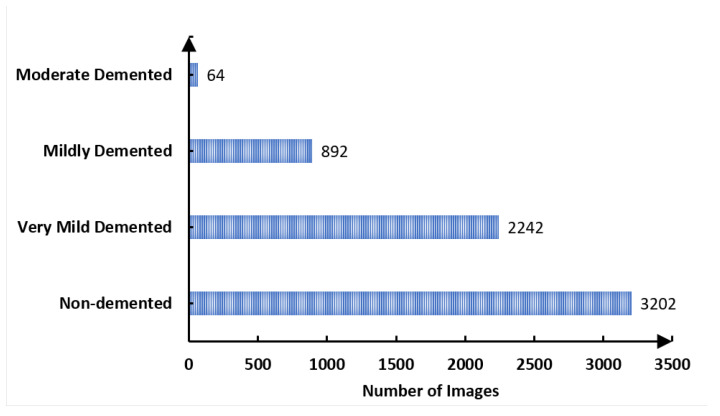
Distribution of our dataset.

**Figure 3 sensors-23-08192-f003:**
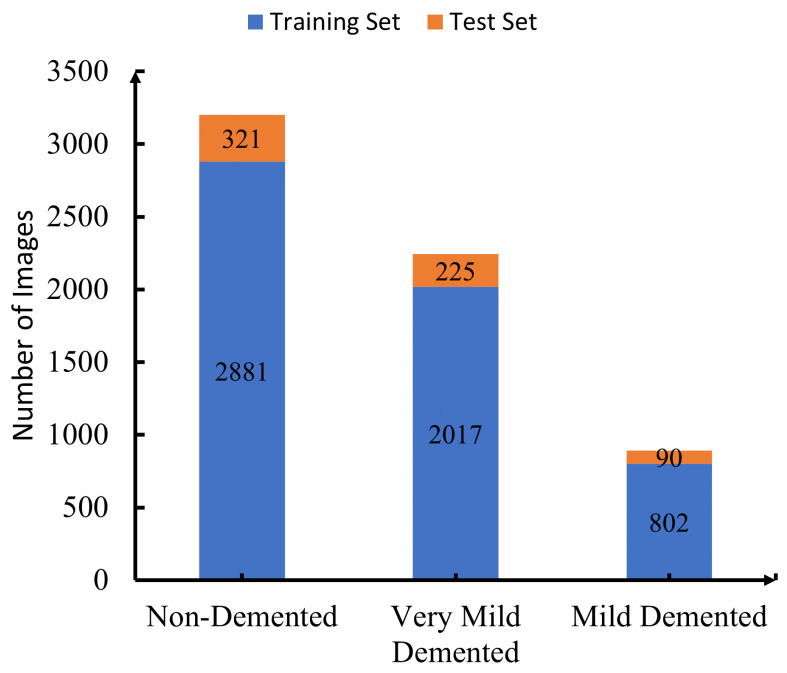
Distribution of test and training dataset.

**Figure 4 sensors-23-08192-f004:**
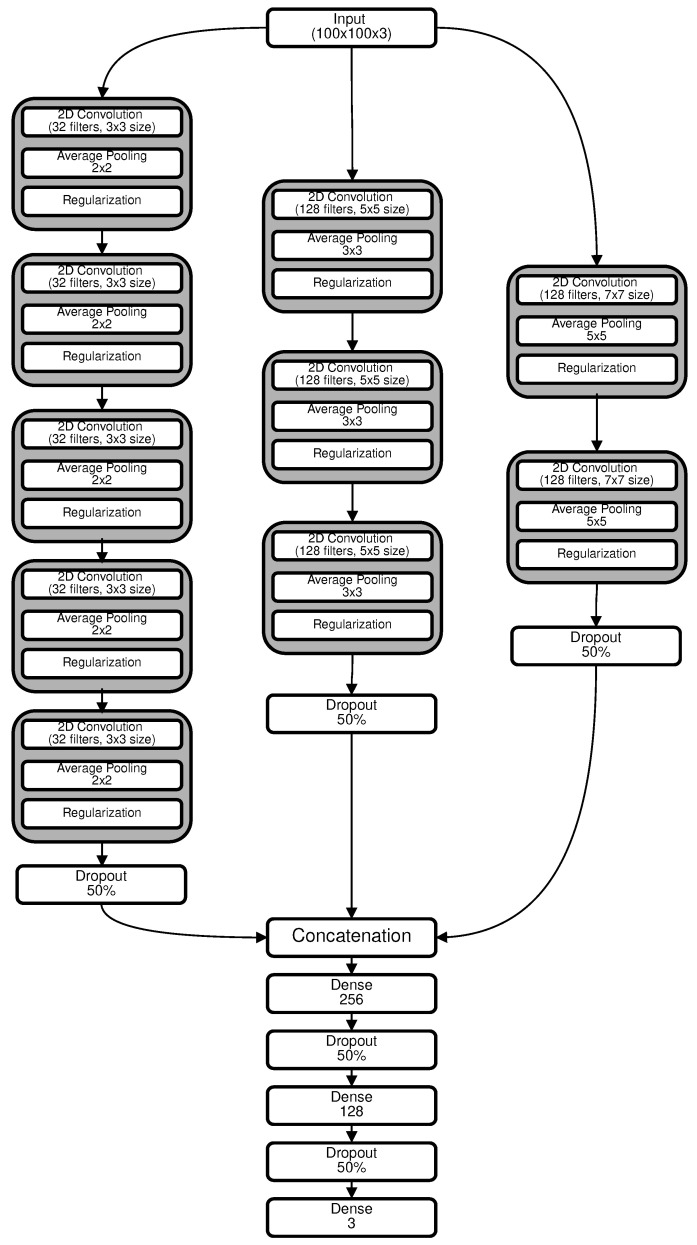
Our proposed network architecture.

**Figure 5 sensors-23-08192-f005:**
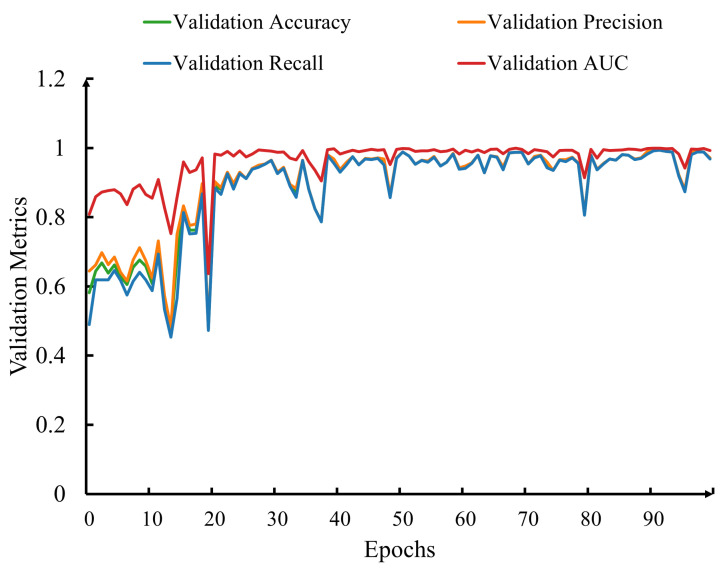
Improvement in accuracy, precision, recall, and AUC against the validation set as the number of epochs increased.

**Figure 6 sensors-23-08192-f006:**
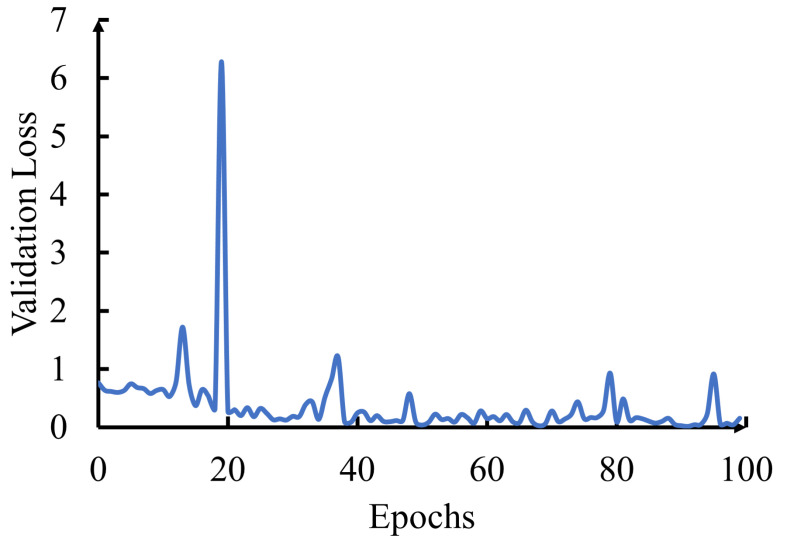
Validation loss evolution over training epochs.

**Figure 7 sensors-23-08192-f007:**
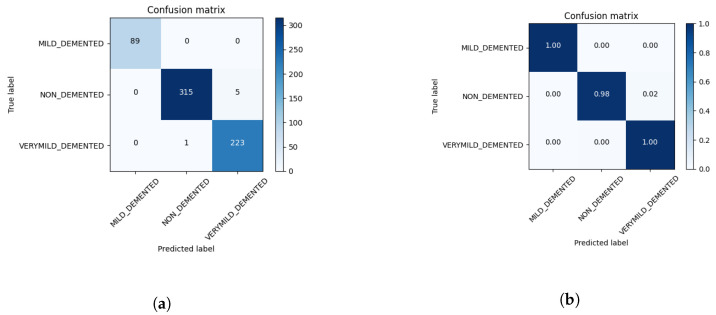
(**a**) Confusion matrix showing the results of our network. (**b**) Normalized confusion matrix.

**Table 1 sensors-23-08192-t001:** Comparative evaluation of AI-enhanced AD diagnosis using MRI images: proposed technique vs. published models.

Reference	Dataset Size	Technique	Performance
Lin W. et al. [10]	725	CNN	79.9%
Bae J.B. et al. [11]	780	CNN	89%
Ebrahimi A. and Luo S. [12]	264	2D CNN and RNN	96.88%
Proposed technique	6388	CNN	99.05%

## Data Availability

Data is publicly available here: https://www.kaggle.com/datasets/tourist55/alzheimers-dataset-4-class-of-images (accessed on 6 October 2022).

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
