# Peer review of "Deep Multi-Branch CNN Architecture for Early Alzheimer’s Detection from Brain MRIs"

_sensors, 2023, doi:10.3390/s23198192_

Round 1
Reviewer 1 Report
The claim of achieving a "99.05% three class accuracy" with the proposed model is ambitious, yet the paper does not provide sufficient evidence of validation, cross-validation, or testing methodologies. The absence of discussions on potential biases, limitations, or generalizability undermines the credibility of this accuracy claim.
In summary, the paper's lack of in-depth discussion, insufficient citation of statistics, vague descriptions of methods and architecture, and absence of rigorous validation procedures result in a work that fails to offer a comprehensive and trustworthy contribution to the field of Alzheimer's disease diagnosis.
Moderate editing of English language required
Author Response
Response to Reviewer 1 Comments
We are sincerely grateful to the reviewer for their valuable comments and insightful suggestions. Below, we address each of the reviewer's comments to further enhance the quality of our work:
Comment 1: The claim of achieving a "99.05% three class accuracy" with the proposed model is ambitious, yet the paper does not provide sufficient evidence of validation, cross-validation, or testing methodologies. The absence of discussions on potential biases, limitations, or generalizability undermines the credibility of this accuracy claim.
Response 1: Thank you for raising a valid criticism of our work. We revised the manuscript to be much more clear about how we used our validation dataset. We apologize for the ambiguity and appreciate your feedback in this regard.
Comment 2: In summary, the paper's lack of in-depth discussion, insufficient citation of statistics, vague descriptions of methods and architecture, and absence of rigorous validation procedures result in a work that fails to offer a comprehensive and trustworthy contribution to the field of Alzheimer's disease diagnosis.
Response 2: We definitely appreciate the reviewer for their criticism. Hence, we have revised the methods and architecture section. We also added the validation procedure for the results obtained in this work.

Reviewer 2 Report
The following aspects must be improved:
-a related work section must be added - including added existing methods with architecture description and obtaining results
-explain the role of sections 1.2-1.5
-section 3 must be removed - it contains only some theoretical aspects, not relevant for the article
-present more clearly the benefits of the proposed architecture
-explain more clearly line 227: we trained on both the training and validation sets and then ran the test set through our neural network - what about validation set; only training and testing sets are presented
-explain more clearly figures 6 and 7 - especially fig. 7 - for what dataset is made?
-refer all figures in the text
Author Response
Response to Reviewer 2 Comments
We extend our heartfelt gratitude to the reviewers for their invaluable comments, insightful suggestions, and constructive feedback, which have significantly contributed to the refinement and enhancement of this paper. Below, we provide a detailed summary of the reviewers' comments and our corresponding responses.
The following aspects must be improved:
Comment 1: a related work section must be added - including added existing methods with architecture description and obtaining results
Response 1: Thank you for making a valuable suggestion. We have added a related work section as per your recommendation.
Comment 2: explain the role of sections 1.2-1.5
Response 2: We sincerely appreciate your feedback. The purpose of sections 1.2-1.5 was to enumerate existing methods. These have now been moved into the prior research section per your request. Your guidance has been immensely helpful in improving the clarity of our manuscript.
Comment 3: section 3 must be removed - it contains only some theoretical aspects, not relevant for the article
Response 3: We appreciate your feedback. After careful consideration, we have opted to retain section 3 in the paper. This section serves a crucial role in providing the necessary theoretical context for our research, and we have also added further justification for its inclusion to ensure its relevance to the article. Your input has contributed to enhancing the overall quality of the paper.
Comment 4: present more clearly the benefits of the proposed architecture
Response 4: We genuinely appreciate your feedback and the opportunity to clarify the benefits of our proposed architecture. To address this, we have incorporated a more explicit description in the introduction section. This enhancement enables readers to discern the novel technique we are presenting in this paper:
"This architectural innovation introduces three distinct convolutional branches, each varying in length and incorporating diverse kernel sizes. The multiple branches with varying kernels allow for better and more nuanced feature extraction as a benefit of the proposed technique. The primary novelty we present in this paper centers on the conception and rigorous evaluation of this intricate CNN model, with the ultimate objective of advancing the accuracy and robustness of AD detection."
Comment 5: explain more clearly line 227: we trained on both the training and validation sets and then ran the test set through our neural network - what about validation set; only training and testing sets are presented
Response 5: We sincerely appreciate your astute observation. In response to your feedback, we have made the necessary modifications to the manuscript by including the validation set.
Comment 6: explain more clearly figures 6 and 7 - especially fig. 7 - for what dataset is made?
Response 6: We are genuinely grateful for your valuable feedback. In our revised manuscript, we have addressed your concerns by providing clearer explanations for figures 6 and 7.
Comment 7: refer all figures in the text
Response 7: Thank you for your thoughtful suggestion. We have incorporated your advice by referring to all figures in the text of the manuscript. Your meticulous review has significantly improved the overall quality of our paper.
In end, we extend our sincere gratitude to you for your dedicated time and insightful comments. Your valuable input has contributed significantly to the clarity and effectiveness of our paper.

Reviewer 3 Report
Based on a review of early detection methods for Alzheimer's disease, this article provides an overview of the dataset from the ADNI and proposes a deep CNN architecture consisting of 7866819 parameters. The model can predict whether patients are non-dementia, mild-dementia, or moderate dementia with a tertiary accuracy of 99.05%. The article is well organized and its presentation is good, which deserve publication in Sensors. Some minor concerns are listed below.
1. Suggest adding a summary statement at the end of the abstract.
2. How does Ref. 13 come after 20 in the text? It is suggested to reorder it.
3. Figure 5, 6, 7 are not mentioned in the main text.
4. Line 274-“The proposed architecture has achieved high accuracy of 99.05% in predicting the progression of the disease and is a step forward in developing tools for”, this sentence is not written completely.
5. There are many errors in the references: a. The format of the reference author is not unified, some use et al, and some are given to all authors. There is also no unified way of writing names; b. Ref. 1,7, 12, 13 etc. does not mark “vol.”; c. No DOI number are given in Ref. 8, 13, 20 etc.; d. Some Ref. year are written before, while others are written after; Ref. 9 and 17 have been officially published and page numbers should be provided.
Minor editing of English language required
Author Response
Response to Reviewer 3 Comments
Comment 1: Based on a review of early detection methods for Alzheimer's disease, this article provides an overview of the dataset from the ADNI and proposes a deep CNN architecture consisting of 7866819 parameters. The model can predict whether patients are non-dementia, mild-dementia, or moderate dementia with a tertiary accuracy of 99.05%. The article is well organized and its presentation is good, which deserve publication in Sensors. Some minor concerns are listed below.
Response 1: Thank you for your time towards the review and for providing constructive feedback.
Comment 2: Suggest adding a summary statement at the end of the abstract.
Response 2: Thank you for the suggestion. We have added the summary statement at the end of the abstract.
Comment 3: How does Ref. 13 come after 20 in the text? It is suggested to reorder it.
Response 3: Thank you again. We have corrected the reference order.
Comment 4: Figures 5, 6, 7 are not mentioned in the main text.
Response 4: Thank you for your observation and valuable comments. As per your comments, we have corrected this issue and added figures 5, 6, and 7 in the main text.
Comment 5: Line 274-“The proposed architecture has achieved high accuracy of 99.05% in predicting the progression of the disease and is a step forward in developing tools for”, this sentence is not written completely.
Response 5: Thank you for pointing out an incomplete sentence in the conclusion section. We have now addressed this issue in the revised manuscript.
Comment 6: There are many errors in the references: a. The format of the reference author is not unified, some use et al, and some are given to all authors. There is also no unified way of writing names; b. Ref. 1,7, 12, 13 etc. does not mark “vol.”; c. No DOI number are given in Ref. 8, 13, 20 etc.; d. Some Ref. year are written before, while others are written after; Ref. 9 and 17 have been officially published and page numbers should be provided.
Response 6: Thank you for helping us improve the reference section of the manuscript. All these references are now fixed.

Reviewer 4 Report
The conducted research provides valuable insights into the application of neural networks for early Alzheimer's Disease (AD) detection. However, there are several weaknesses and areas that require further consideration to strengthen the conducted research:
1: First and foremost, the dataset is imaged from other sources as mentioned in the paper, which may not fully represent the diversity of AD cases or variations in imaging which could potentially limit the generalizability of the model to broader populations.
2: Also, the conducted research does not clearly indicate how the model's predictions are into line with clinical diagnoses. Likewise, the sample size is also small and the designed model could easily fail on real-time data.
3: In the conducted research the authors do not discuss or prevention of over-fitting or underfitting. Hence, the generalizability of the model is not explained.
4: The research conducted is so simple and there has been a lot of such research conducted in the past with a far better approach with respect to the implementation and novelty of AI models. For instance, there is a need to explore and incorporate additional features, such as demographic data or other biomarkers. Just using a simple CNN for classification is outdated and simple and lacks novelty.
5: Just using the simple neural networks on limited-on-limited data related to AD diagnosis can create severe ethical and reliable considerations.
6: The conducted research also lacks external validation, hyperoperators, model interpretability and expandability.
7: For the future submission, the authors need to improve the introduction, methodology, experimental results and comparison, provide tradeoffs of their conducted research, and clinical significance.
Based on the provided weakness, we made the decision that this research is too simple and lacks novelty. Therefore, we decline this research to be accepted as per the quality requirements of this journal.
NA
Author Response
Response to Reviewer 4 Comments
We would like to express our sincere gratitude to the reviewer for dedicating their time and expertise to review our manuscript. We greatly appreciate your thoughtful comments and valuable suggestions, which have significantly contributed to the refinement of our work. Below, we provide responses to your insightful comments and hope that our revisions adequately address your concerns.
The conducted research provides valuable insights into the application of neural networks for early Alzheimer's Disease (AD) detection. However, there are several weaknesses and areas that require further consideration to strengthen the conducted research:
Comment 1: First and foremost, the dataset is imaged from other sources as mentioned in the paper, which may not fully represent the diversity of AD cases or variations in imaging which could potentially limit the generalizability of the model to broader populations.
Response 1: We sincerely appreciate your thoughtful consideration of our dataset. As you mentioned, other MDPI papers have also utilized this Kaggle dataset [link to the paper: https://www.mdpi.com/2075-4418/13/7/1216]. Furthermore, the dataset itself explicitly states its origin as Kaggle. We understand your concern about the dataset's representativeness and potential limitations in generalizability, and we will ensure that this aspect is duly acknowledged in our paper. Your insights are invaluable and contribute to a more comprehensive understanding of our research.
Comment 2: Also, the conducted research does not clearly indicate how the model's predictions are into line with clinical diagnoses. Likewise, the sample size is also small and the designed model could easily fail on real-time data.
Response 2: We genuinely appreciate your insights and concerns. To address these valid points, we would like to highlight that our dataset, as shown in Table 1, comprises 6400 images, notably more extensive than many other published research studies in this field. This substantial dataset size allows us to provide a comprehensive analysis.
Comment 3: In the conducted research the authors do not discuss or prevention of over-fitting or underfitting. Hence, the generalizability of the model is not explained.
Response 3: We apologize for any confusion related to the validation set, and we have made substantial improvements to clarify this aspect.
Comment 4: The research conducted is so simple and there has been a lot of such research conducted in the past with a far better approach with respect to the implementation and novelty of AI models. For instance, there is a need to explore and incorporate additional features, such as demographic data or other biomarkers. Just using a simple CNN for classification is outdated and simple and lacks novelty.
Response 4: We appreciate your perspective on our research. At the time when we concluded our experiments, we achieved the highest reported accuracy in Alzheimer's detection. While we understand the importance of exploring and incorporating additional features, such as demographic data or other biomarkers, it's essential to note that our study is constrained by the data available to us for experimentation. We acknowledge the value of these additional factors and their potential impact on model performance. However, we are limited to the data provided by ADNI, which includes demographic data but has a significantly smaller number of MRI images. We appreciate your feedback and insights, which contribute to a more comprehensive understanding of the scope and limitations of our research.
Comment 5: Just using the simple neural networks on limited-on-limited data related to AD diagnosis can create severe ethical and reliable considerations.
Response 5: We appreciate your concerns regarding the ethical and reliability considerations associated with our research. It's important to note that our model is trained on the largest set of publicly available MRI imaging data for AD diagnosis. We have taken into account the potential reliability concerns, as discussed in lines 353-359 and 371-381 of our manuscript. We understand the importance of addressing ethical and reliability considerations in our work and will continue to prioritize these aspects in our research.
Comment 6: The conducted research also lacks external validation, hyperoperators, model interpretability and expandability.
Response 6: We have elaborated significantly on how we selected our validation dataset and tuned hyper-parameters. Additionally, neural networks have notoriously low interperatibility due to the complexity of the features that are extracted. Thus, model interpretability is well outside the scope of this paper.
Comment 7: For the future submission, the authors need to improve the introduction, methodology, experimental results and comparison, provide tradeoffs of their conducted research, and clinical significance.
Response 7: We highly appreciate your feedback and constructive suggestions for improvement. In response to your valuable comments, we have made substantial revisions to enhance our manuscript's introduction, methodology, experimental results, and comparison sections. Additionally, we have worked to provide a clearer discussion of the trade-offs in our research and its clinical significance.
In the end, we sincerely appreciate your valuable comments and suggestions. Your insightful input has played a pivotal role in enhancing the quality of our paper, and we are genuinely grateful for your contributions.

Round 2
Reviewer 2 Report
Some of my comments were addressed.
In my opinion section 4 contains only some common theoretical aspects that are not relevant for the paper and this must be removed.
For a better evaluation, confusion matrix must be normalised.
Figure 6 is still not referred in the text.
Author Response
Response to Reviewer 2 Comments
We would like to express our sincere appreciation for your thorough review of our paper and for your valuable feedback. Your input has been instrumental in improving the quality of our work. We have carefully considered your comments and are grateful for your insights.
Some of my comments were addressed.
Comment 1: In my opinion section 4 contains only some common theoretical aspects that are not relevant for the paper and this must be removed.
Response 1: Thank you for your feedback. Based on your suggestion, we have removed Section 4 from the paper and integrated the relevant content from that section into the new Section 4.3.2. This adjustment allows us to streamline the paper and maintain its focus on the core themes. We appreciate your guidance in enhancing the paper's clarity and relevance.
Comment 2: For a better evaluation, confusion matrix must be normalised.
Response 2: Thank you for your valuable suggestion. Per your recommendation, we have incorporated the normalized confusion matrix into Figure 7b. Additionally, we have included a reference to this figure in the text to provide better clarity and context for our readers. Your input has helped improve our paper’s visual representation and overall comprehensibility. We appreciate your thoughtful guidance.
Comment 3: Figure 6 is still not referred in the text.
Response 3: We sincerely apologize for not being able to address your previous comment adequately. We have now referred to Figure 6 (now Figure 4) in the revised manuscript as per your last suggestion. We value your feedback, and guidance has been essential in refining our paper. Thank you for your patience and valuable input.
Once again, we sincerely appreciate your thoughtful review and commitment to improving the quality of our work. Your feedback has been invaluable in refining and enhancing our paper. Thank you for your time and expertise.

Reviewer 4 Report
All the concerns raised during our previous review have been addressed adequately. Therefore, we consider this paper for possible publication (subject to EiC decision).
NA
Author Response
Response to Reviewer 4 Comments
Comment 1: All the concerns raised during our previous review have been addressed adequately. Therefore, we consider this paper for possible publication (subject to EiC decision).
Response 1: We are genuinely grateful for your positive assessment of the revisions we have made to our paper based on your previous comments. Your recognition that we have effectively addressed all concerns is both motivating and reassuring.
Your feedback and meticulous review have undeniably contributed to enhancing our work, aligning it with the high standards of scientific excellence expected by the journal. We remain fully committed to upholding these standards throughout the publication process.
We sincerely appreciate your dedicated time and expertise in evaluating our paper. Your valuable input has been instrumental in its improvement.
